# Effects of Herd Establishment Time and Structure on Group-on-Individual Aggression Intensity in Farm Pigs

**DOI:** 10.3390/ani14152229

**Published:** 2024-07-31

**Authors:** Zhen Wang, Zhengxiang Shi, Hao Li, Hui Liu, Zhaowei Xiao, Hao Wang, Shihua Pu

**Affiliations:** 1College of Water Resources & Civil Engineering, China Agricultural University, Beijing 100083, China; wang@cau.edu.cn (Z.W.); leehcn@hotmail.com (H.L.); liuh@cau.edu.cn (H.L.); sy20233092119@cau.edu.cn (Z.X.); 2Chongqing Academy of Animal Sciences, Chongqing 402460, China; wanghaocau@163.com (H.W.); push@cqaa.cn (S.P.); 3National Center of Technology Innovation for Pigs, Chongqing 402460, China

**Keywords:** animal welfare, aggression, herd structure, herd establishment time, pig mixing

## Abstract

**Simple Summary:**

Individuals integrated into a new social group may suffer from group aggression, which is common in animals but remains unresolved. The objective of this study was to investigate the effect of the establishment time and structural composition of different herds on the intensity of aggression toward unfamiliar pigs. Six pig herds were established, with a pig integrated into each herd at intervals of 3, 6, 9, 12, 15, 18, and 21 days after their establishment. A parabolic model based on the Levenberg–Marquardt algorithm and conjoint analysis identified factors influencing the intensity of aggression toward unfamiliar pigs. Our results indicate that aggression intensity towards unfamiliar pigs in the herd was not significantly influenced by herd size but was significantly affected by the number of pens used to form the herds. Furthermore, we found a significant and positive correlation between herd establishment time and aggression intensity, which was consistently observed across different pig herds. The impact of the number of pens on aggression intensity showed a significant correlation with the interaction effect of the number of pens and establishment time. These findings underscore the complexity of interactions between herd establishment time and structural composition in influencing aggression intensity towards unfamiliar pigs.

**Abstract:**

Aggression in farm animals affects welfare. Although one-on-one aggression was studied, group-on-individual aggression remains unresolved. This study aimed to examine how herd establishment times and structures influence aggression intensity (AI) of herds towards unfamiliar pigs. Six groups of pigs were established, with a new pig added every three days. AI was measured by skin lesion severity on the new pigs. A parabolic model based on the Levenberg–Marquardt algorithm and conjoint analysis identified factors influencing AI. Results show AI was not significantly affected by herd size but was significantly influenced by the number of pens (*p* < 0.01). AI showed a significant association with elevated time (T) since the establishment of the herd in six pig herds (Kendall’s tau-τ = 0.976, *p* < 0.001). The effect of T on the AI became stronger as T increased, which was consistently validated in six pig herds. Furthermore, the interaction effect indicates a significant difference in AI between herds formed with two pens and those with more than two pens when T ≤ 12 (*p* < 0.05). However, as T increased beyond 12, the number of pens used to form the herd did not significantly affect AI. These findings highlight the complex interactions between herd establishment time and structural composition in shaping aggression intensity towards unfamiliar pigs.

## 1. Introduction

Globally, 80 billion land animals are reared for human consumption each year, more than 75 billion of which are farm animals [1,2]. As this number continues to rise, public concern for the welfare of farm animals is also growing. In response to this trend, legislative reforms and industry initiatives increasingly emphasize the improvement of farm animal welfare standards [3]. This includes a growing adoption of group housing systems for sows over traditional individual confinement stalls, aligning with sow biology to minimize stress and ensure animal welfare [4,5,6].

Based on the biological characteristics of sows needing individual housing during the implantation and lactation periods post-mating, and group housing at other times, they experience at least two transitions from individual to group housing within one reproductive cycle [7,8,9]. However, in the mixing process from individual to group housing, due to individual differences, sows may miss the collective mixing time due to early or delayed weaning, resulting in the emergence of solitary individuals. Furthermore, sows identified as non-pregnant post-mating, experiencing miscarriage, or requiring isolation due to illness are temporarily separated from the group, which also leads to the emergence of solitary individuals [10,11,12]. Managing these separated individuals individually impacts economic efficiency and perpetuates individual confinement practices. Conversely, due to differing growth stages, varied times of separation, and behavioral differences, mixing them into one group is impractical. Therefore, it is usually necessary to integrate the solitary individual into a pig group that already existed, either their original group or a new one. Even if separated individuals can return to their original group, based on the reproductive and physiological characteristics of sows, this occurs after 3–21 days of separation [13]. Previous studies showed that individuals, after being separated from the group for a period of time, even upon returning to their original group, still need to re-establish hierarchical relationships with other members [14]. This implies that upon returning to the original group, individuals are nearly akin to newcomers to the group. During the process of integrating into a new social group, individuals often face attacks from dominant subgroups composed of the resident members [15,16,17]. This not only increases the risk of injury for individuals, but also places them under higher stress and social instability [18]. Moreover, these newly integrated individuals, due to their captive environment, cannot escape group attacks, often resulting in them being bitten, maimed, or even killed. This not only affects their welfare and psychological well-being, but also negatively impacts the stability and productivity of the entire group [19,20]. Therefore, understanding which groups are more likely to accept new individuals and taking measures to reduce occurrences of aggressive behavior is crucial for improving animal welfare and optimizing farming management when handling these separated solitary individuals. However, the types of groups with lower intensity of aggression towards unfamiliar individuals and the reasons behind this phenomenon remain unknown.

While there was extensive research on the transition of sows from individual housing to group housing systems and the aggression issues it triggers [21], the problems of aggression that may arise when isolated solitary individuals integrate into an existing group remain unresolved. Detailed research on this process remains limited, particularly in understanding the dynamics of group aggression towards newly integrated individuals and the controlling factors. Given the inherent differences between simultaneous mixing of unfamiliar individuals and the integration of solitary unfamiliar individuals into established groups (Figure 1), a pilot study was conducted to provide preliminary observations on factors influencing aggression intensity towards unfamiliar pigs by the group. Based on these preliminary observations, hypotheses were proposed that the time (T) since the establishment of the herd and the herd’s structural composition (including size and the number of pens) would affect the aggression intensity (AI) of the herd toward an unfamiliar pig, with potential interaction effects among these influencing factors.

To test this hypothesis, the study simulated the integration of individual first-parity sows weighing 150 kg, as they constitute a larger proportion within sow groups. This corresponds to higher numbers of sows experiencing failed insemination and abortion, indicating a greater need for integration. The research investigated how the establishment time and structural composition of different herds influenced aggression levels toward unfamiliar pigs. By providing empirical data, the study aims to enhance the welfare of individual pigs during their integration into new social groups in practical production settings. Furthermore, it addresses public concerns regarding farm animal welfare, potentially informing future agricultural policies and industry standards.

## 2. Materials and Methods

### 2.1. Pig Herd and Rearing Conditions

This study was conducted at a traditional fattening pig farm located in northeastern China (45°49′ N, 130°34′ E) over multiple batches from 2021 to 2023. This is an intensive closed-cycle (birth-to-market) farm, meaning that it produces piglets and raises them to market weight. Such mixed-type small-scale farms are common in China and typically feature some arable land used for growing energy-intensive crops (most commonly corn). Consequently, the animals are partly fed home-grown crops supplemented with commercial feed and supplements. Depending on their growth stage, pigs are housed in indoor systems. Sows during lactation are kept in farrowing crates to limit their movement and reduce the risk of injury to newborn piglets. Piglets are transferred to nursery pens at 4–6 weeks of age, and then to finishing pens at 11 weeks, until approximately 36 weeks when they reach a weight of 160 kg and are sold (differing from traditional commercial pig farms in terms of breeding duration for producing green pork). However, the sows leave the farrowing room 7 days earlier than piglets and are transferred to the sow barn. The sow barn (length × width × height: 23 × 11 × 2.3 m) consists of multiple double-row units facing each other. Each unit can house up to 4 sows or up to 6 first-parity sows and near-first-parity sows. Sows remain in the sow barn during their open period, breeding, and early pregnancy. After 21–35 days of pregnancy, sows are moved to the group housing sow barn (length × width × height: 12 × 6 × 2.3 m), which is equipped with movable partitions to accommodate different groupings based on parity, weight, and other conditions.

To ensure adequate water intake, each pen was equipped with automatic waterers. Nutritionally, all pigs were fed a diet composed and balanced according to the recommendations of the NRC (2012), including yellow corn, wheat bran, soybean meal, corn oil, calcium carbonate, dicalcium phosphate, salt, L-lysine hydrochloride, L-threonine, L-tryptophan, DL-methionine, and vitamins [22]. The daily feed quantity was determined based on the weight and growth stage of each pig to ensure sufficient nutrient intake. The feeds were accurately weighed and adjusted according to the feeding records to meet the growth requirements of each pig. Feeding is scheduled to maintain feed quality and nutritional integrity. Furthermore, multiple measures are taken to ensure feed freshness, including selecting high-quality ingredients, strictly mixing according to formulation proportions, regularly inspecting feed for mold or bacterial contamination, and maintaining a dry and clean feed storage environment. To avoid any potential impact on pig aggression, we did not use ractopamine or other growth-promoting additives. Moreover, the pigs were provided with roughage (straw and cabbage) daily at 13:00 to increase oral activity. Throughout the trial, stringent environmental controls were implemented to maintain optimal pen conditions, with the temperature regulated between 18 and 20 °C and relative humidity maintained between 65% and 70%; the pigs were checked at least once daily to ensure free access to water and maintain good health. Additionally, all the experimental procedures were approved by the China Agricultural University Laboratory Animal Welfare and Animal Experiments Ethics Committee according to the China Animal Protection Ordinance (authorization number AW70803202-5-2).

### 2.2. Methods for Identifying Aggressive Behavior and Quantifying Its Intensity

Given the variety of aggressive behaviors due to species differences, there are dif-ferences between species in the methods used to quantify aggression intensity. Prior to the commencement of the main experiment, an additional 37 pigs were used for a pilot study to provide preliminary observation results. The pilot test is presented in the Appendix A.

In the pilot test, we found that there were four main types of aggressive behavior under the conditions of integrating a pig into a new herd: biting, touching, sniffing, and looking. Although the latter three behaviors were aggressive in this case, they did not cause lesions to the skin of the pigs. As a result, this study identified only biting as aggressive behavior. In the initial stages of integration, we found that integrated pigs rarely initiated aggression and mostly became bitten pigs, while resident pigs usually initiated aggression and were directed toward integrated pigs (this phenomenon was also confirmed in subsequent main experiments), implying that the aggression intensity of the (resident) herd was mainly reflected in the degree of skin lesions caused by resident pigs to integrated pigs, thus allowing us to indirectly quantify AI by means of skin lesion scoring of the integrated pig. The classification of skin lesion levels and their definitions in integrated pigs are shown in Appendix A. In subsequent main experiments, we further refined and detailed the quantification methods and evaluation criteria for aggression intensity (see Section 2.5).

### 2.3. Experimental Hypothesis and Protocol

Furthermore, we also found that the number of resident pigs involved in aggression in the herd varied with T and that the aggression intensity of the resident pigs varied with the number of pigs from different pens. Accordingly, we hypothesized that AI would be related to T and the structural composition of the herd (including size and the number of pens). Subsequently, we designed two series of experiments, Experiment 1 and Experiment 2, to test the hypothesis.

Experiments 1 and 2 were conducted in a group-housing sow barn, involving a total of 102 primiparous sows (35 Landrace and 67 Large White × Landrace), with pregnancies of 35 days or more, each weighing approximately 150 kg. The experiments selected first-parity sows weighing 150 kg due to their higher proportion in sow groups, which corresponds to a higher number of sows experiencing failed insemination and abortion, indicating a greater need for integration.

In Experiment 1, we investigated the impact of herd size and establishment time on the intensity of aggression toward newly integrated unfamiliar pigs. We formed 4 groups, including two control groups and two treatment groups. Control 1 (CTRL1) consisted of 9 primiparous sows randomly selected from two pens in the sow barn (5 pigs from one pen and 4 pigs from another pen), and Treatment 1 (TREAT1) consisted of 12 primiparous sows randomly selected from two pens in the sow barn (6 pigs from each pen). Control 2 (CTRL2) also included 9 primiparous sows randomly selected from two pens in the sow barn (5 pigs from one pen and 4 pigs from another pen). Treatment 2 (TREAT2) consisted of 12 sows randomly selected from two units in the sow barn (6 pigs from each pen). Afterward, T was divided into 7 intervals for the aforementioned groups, with each interval lasting 3 days. The intervals began at T = 3 as the initial value and concluded at T = 21 as the cutoff. Unfamiliar pigs were integrated into each herd at T = 3, 6, 9, 12, 15, 18, and 21. To ensure accurate measurement of aggressive behavior and minimize interference in determining the effects on pig group dynamics, we chose to integrate only one pig into each group at each time point (Figure 2).

In Experiment 2, we investigated the impact of the number of pens composing the herds and herd establishment time on the intensity of aggression toward newly integrated unfamiliar pigs. There were 4 groups in Experiment 2, including 2 groups from Experiment 1. These comprised 2 control groups and 2 treatment groups. The control groups (CTRL1 and CTRL2) each consisted of 9 primiparous sows randomly selected from two pens in the sow barn (5 pigs from one pen and 4 pigs from another pen). The treatment groups (TREAT3 and TREAT4) each consisted of 9 primiparous sows randomly selected from more than two pens in the sow barn: 4 pens (numbers of pigs per pen: 2, 2, 2, and 3, respectively) and 6 pens (numbers of pigs per pen: 2, 2, 2, 1, 1, and 1, respectively). At each of 7 time points (T = 3, 6, 9, 12, 15, 18, and 21), unfamiliar pigs were integrated into each herd. It was the same as the Experiment 1 procedures, only one pig was integrated into each group at each time point.

Experiment 1 and Experiment 2 were performed for 42 integrated pigs and 6 pig herds in total. For each series of experiments, 7 pigs were integrated into the same herd at 7 different times. Each integrated pig was unfamiliar with the resident pigs, while its breed was consistent with that of the resident pigs. Familiarity at this stage was defined as sharing a previous pen with no regard to whether they were littermates. For each integration, a professional breeder and a professional veterinarian were involved in addition to the experimenters conducting the study. Based on actual production experience, breeders integrated unfamiliar pigs into these herds without violating the guidelines for the welfare of commodity pigs. In accordance with the biosafety requirements of pig farms, veterinarians disinfected and treated pigs with moderate and worse skin lesions.

### 2.4. Behavior Data Collection

Based on animal behavioral research methods [23,24], integrated pigs were initially observed by the human eye from the beginning of integration to the end of integration, an integration ethogram was recorded, and at the end of each integration, skin lesions on the integrated pigs were captured by a camera. In each integration, the end of the integration was signaled when the resident pigs stopped paying excessive attention to and initiating biting toward the integrated pigs or when there were no new skin lesions on the integrated pigs. A new skin lesion was considered if it exhibited vivid red coloration. Due to differences in the T and structural compositions of the pig herds, the AI varied, resulting in different times from the beginning to the end of each integration. Integration typically occurs after resident pigs and integrated pigs completed three feeding and sleeping cycles together, although the duration might vary. As a result, the behavioral observation time for each integration varied by herd, but at the end of each integration, one photograph of skin lesions on integrated pigs was captured. However, in special cases, such as when the level of skin lesions on integrated pigs reached the highest level defined in Table 1 (serious levels) at a certain T, it is reasonable to predict that the level of skin lesions on integrated pigs may be more serious at the next T, which could put their lives at risk. Therefore, when the level of skin lesions in the integrated pigs reached a serious level, no more pigs were integrated into the herd, and accordingly, no photographs were taken. This situation occurred only once in this study, specifically in TREAT1, when T = 21. Experimentally captured photographs are presented in Appendix A.

Experimental data primarily recorded skin injuries from newly integrated pigs, rather than skin injuries from all pigs within the new social group. The main reason for this is that during the experiment, there was only one instance where an original resident pig was lightly bitten by a newly integrated pig, specifically in TREAT4 when T = 3, which was recorded in Appendix A. Apart from this, there were no recorded skin injuries caused by newly integrated pigs among the residents. This phenomenon arises because newly integrated pigs seldom initiate aggression and often experience mobbing before they can select their targets. They occasionally retaliate, but such retaliations do not effectively harm the original resident group. Similar behaviors were observed in studies of group integration in mice and chimpanzees [25,26]. Newly integrated members lack the ability to effectively challenge the absolute dominance of the original resident group [27,28]. Any resistance they show is typically a brief instinctual response aimed at avoiding further aggression [29].

### 2.5. Classification and Assessment Method of Skin Lesions

In this study, we chose lesion intensity as the evaluation metric because it is easier to observe, record, and quantify, and it effectively reflects aggression intensity. During the experiment, we found that the duration of attacks and the number of injuries did not accurately reflect aggression intensity. For example, both biting and ramming are forms of aggression, a single attack can cause multiple injuries, and dozens of minor injuries may not be as severe as a single major injury. Additionally, the use of lesion intensity as a quantitative measure of aggression was well validated in previous studies [30,31].

Skin lesions, which serve as indicators of aggressive interactions [32], were assessed before integration and upon completion of each integration by a single trained observer. The preintegration skin lesion count was subtracted from that at the end of each integration for each integrated pig and resident pig involved in aggression. This approach ensured that only those lesions that occurred as a result of integration-induced aggression were included in all analyses. According to the findings from the pilot test (Appendix A), the skin lesions of the integrated pigs were classified into six levels based on their severity. The details of these levels and their respective assessment methods are provided in Appendix A. Considering instances where integrated pigs were not bitten by the resident herd or actively bit the resident pigs, we expanded the classification of skin lesions to include these scenarios. Consequently, recently developed skin lesions were further delineated into eight levels, distinguishing between different grades based on the extent of injury (refer to Appendix A and Table 1 for specifics). A skin lesion was considered recent if it exhibited vivid red coloration or had recently formed scabs.

### 2.6. Assessment of Aggression Intensity and Score Allocation

In the assessment of aggression intensity, skin lesion scores served as the primary evaluation metric. Each integrated pig was assigned an aggression intensity score based on the severity of its skin lesions. These scores were calculated by subtracting the preintegration lesion count from the postintegration count for both integrated and resident pigs involved in aggression, ensuring that only lesions resulting from integration-induced aggression were considered. The methodology for assigning AI scores was structured as follows: (1) Preintegration and Postintegration Lesion Counts: skin lesions on integrated pigs were counted before integration and after each integration, capturing any changes resulting from the integration process. (2) Calculation of AI Scores: the difference between postintegration and preintegration lesion counts was calculated for each integrated pig and resident pig involved in aggression, reflecting lesions specifically attributable to integration-induced aggression. (3) Scoring Criteria: AI scores ranging from −1 to 6 were assigned based on the severity of the observed lesions, as outlined in Table 1. These scores corresponded to different levels of aggression intensity, with higher scores indicating more severe aggression-induced lesions. (4) Photographic Documentation: Photographs were taken at the end of each integration to visually document the extent of skin lesions on integrated pigs, providing additional support for the assigned AI scores. The resulting AI scores offered a quantitative measure of the aggression of the resident pigs towards integrated pigs within each herd. These scores served as a reliable evaluation metric for assessing the impact of integration on pig welfare and behavior. The detailed results of the AI scoring process are provided in Appendix A.

### 2.7. Aggression Intensity Analysis Method and Evaluation Index

#### 2.7.1. Levenberg–Marquardt Algorithm

The Levenberg–Marquardt (LM) algorithm is a method for solving nonlinear least squares problems. Unlike the Gauss–Newton algorithm, the LM algorithm can overcome the limitation of having full-rank columns in the matrix during iteration, thus expanding its applicability [33,34]. This algorithm obtains the search direction by solving the following optimization model:$$d_{k}=\arg\underset{d\in R^{n}}{min}{∥J_{k}d+r_{k}∥}^{2}+\mu_{k}{∥d∥}^{2}$$
where $\mu_{k}>0$ and $d_{k}$ are satisfied by the optimality condition:$$\left(J_{k}^{T}J_{k}+\mu_{k}I\right)d_{k}+J_{k}^{T}r_{k}=0$$
$$d_{k}=-{\left(J_{k}^{T}J_{k}+\mu_{k}I\right)}^{-1}J_{k}^{T}r\left(x_{k}\right).$$

The LM algorithm is essentially a combination of the Gauss–Newton algorithm and the trust region method for solving constrained optimization problems. According to linear algebra, the matrix ${\left(J_{k}^{T}J_{k}+\mu_{k}I\right)}^{-1}$ acts on the vector $J_{k}^{T}r\left(x_{k}\right)$ by changing its length and direction. Hence, $∥d_{k}\left(\mu \right)∥$ monotonically decreases with *μ* > 0 and tends to zero as *μ* approaches infinity. Compared to the Gauss–Newton algorithm, the LM algorithm introduces a positive parameter *u* to prevent large changes in the search direction when the matrix $J_{k}^{T}J_{k}$ is close to singular, thus improving the stability.

We provide further clarification on our decision to use the LM algorithm. First, our study addresses complex nonlinear structures and diverse influencing factors, which may pose challenges for traditional linear mixed models in capturing such intricacies within the data. In contrast, the LM algorithm excels in nonlinear optimization tasks, and is particularly adept at handling multivariate nonlinear relationships present in our dataset. Second, social network analysis typically focuses on describing and analyzing intricate interactions among individuals or groups, whereas our research emphasizes potential nonlinear dynamics and response patterns in animal behavior. This characteristic renders the LM algorithm particularly suitable due to its focus on parameter optimization to enhance model–data fit. Highlighting the advantages of the LM algorithm over mixed models, we emphasize its robustness and convergence properties in tackling complex optimization challenges. Parameterization, such as the introduction of the μ parameter, enhances sensitivity to data characteristics, ensuring reliability and precision in addressing diverse optimization tasks. Last, the widespread application and rigorous validation of the LM algorithm in engineering and scientific domains further underscore its suitability for addressing the specific analytical demands of our study. While mixed models offer flexibility in certain contexts, we contend that the LM algorithm provides a more direct and effective approach tailored to the complexities inherent in animal behavior research. In summary, the LM algorithm, as a nonlinear optimization tool, emerges as our preferred choice due to its capability to accurately elucidate key dynamics and response patterns in animal behavior studies. We are confident that this methodological approach will facilitate a more precise and insightful analysis of our research objectives.

#### 2.7.2. Parabolic Model Based on the Levenberg–Marquardt Algorithm

The causes of differences in the aggression intensity (AI) of the herd toward an unfamiliar pig across pig herds were analyzed using a parabola model based on the LM method above. In the model, the time (T) since the establishment of the herd was used as the predictor variable, and the AI score was used as the response variable.

$J_{k}$ describes the derivative of the error function with respect to model parameters in the LM algorithm, specifically corresponding to the derivative of the AI score with respect to the T, aimed at understanding its rate of change. The regularization parameter $\mu_{k}$ influences the step size during the optimization process. $\mu_{k}$ can be understood as a tool to adjust model complexity or control the fit of data, aiming to better fit the AI score. $r_{k}$ represents the difference between actual observed values and model predictions, specifically corresponding to the residuals between the observed AI score in different groups and the model predictions. $d_{k}$ denotes the search direction during the optimization process, adjusting parabolic model parameters by minimizing the error function and regularization term. This explains how the LM algorithm optimizes the prediction of the AI score in different groups by adjusting parabolic model parameters.

The root mean squared error (*RMSE*) was used to assess the precision of the parabolic model in predicting AI scores, measuring the degree of mean deviation between predicted and true values [35]. A lower *RMSE* value indicates smaller prediction errors of the model, thereby suggesting more reliable predictions. The root mean square of the error, or the standard deviation of the residuals, is equal to the square root of the reduced chi-square:$$RMSE=\sqrt{Reduced\chi^{2}}$$
where $Reduced\chi^{2}$ is the reduced chi-square value, which equals the residual sum of squares divided by the degree of freedom:$$Reduced\chi^{2}=\frac{\chi^{2}}{{df}_{Error}}=\frac{RSS}{{df}_{Error}}$$
where *RSS* is the residual sum of squares.

Adjusted *R*-square is used to measure the goodness of fit of the parabolic model to the observed data, considering the number of variables used in the model [36]. In our study, adjusted *R*-square assesses the extent to which the T explains the variability in AI scores across CTRL1, TREAT1, CTRL2, TREAT2, TREAT3, and TREAT4 groups. A higher adjusted *R*-square indicates that the model can better explain the variability of the dependent variable. The adjusted *R^2^* value:$${\bar{R}}^{2}=1-\frac{\frac{RSS}{df_{Error}}}{\frac{TSS}{df_{Total}}}$$
where ${\bar{R}}^{2}$ is the adjusted *R^2^* value and *TSS* is the total sum of squares.

*R* is used to reflect the correlation between predicted values by the parabolic model and actual observed values. In our study, the magnitude of *R* indicates the accuracy of the parabolic model in predicting AI scores. A high correlation coefficient (*R* close to 1) indicates a strong correlation between model predictions and actual observations. The *R* value is the square root of *R*^2^:$$R^{2}=\frac{Explainedvariation}{Totalvariation}=\frac{TSS-RSS}{TSS}=1-\frac{RSS}{TSS}$$
$$R=\sqrt{R^{2}}.$$

Hence, *RMSE*, adjusted *R*-square, and *R*, together with the parameters of the LM algorithm ($J_{k}$, $u_{k}$, $r_{k}$, and $d_{k}$), constitute the evaluation framework of our study’s model. Through a comprehensive analysis of these metrics, we assess the model’s ability to explain the effect of time since herd establishment on aggression intensity scores towards unfamiliar pigs across various pig herds, as well as its prediction accuracy and overall model fit.

#### 2.7.3. Conjoint Analysis Model

Additionally, we used a conjoint analysis model to analyze all of the factors hypothesized to affect AI in this study and evaluated the relative contribution of each factor through its utility value and relative importance [37]. In the conjoint analysis model, the first level of each influence factor is used as the reference term, followed by the ordinary least squares. After obtaining the regression coefficient value for each level, the utility value of the reference item is calculated in turn; the greater the utility value is, the greater the contribution rate. The relative contribution rate of each influencing factor was calculated using the steepest descent method. This includes subtracting the minimum utility value of the influencing factor from the maximum utility value of the corresponding level of the influencing factor. Then, the relative contribution rate is normalized, and the relative contribution rate of each influencing factor is obtained [38].

### 2.8. Statistical Analysis

All statistical analyses were conducted using SPSS AU 2024 (SPSSAU.COM). Schematic diagrams were created using Microsoft Visio 2010 software (Microsoft Corporation, Redmond, WA, USA), and data plots were generated using Origin 2018 software (OriginLab Corporation, Northampton, MA, USA).

Prior to selecting data analysis methods, a normality assessment was performed to confirm adherence to the assumption of normal distribution. Shapiro–Wilk and Jarque-Bera tests indicated that all six datasets (CTRL1, TREAT1, CTRL2, TREAT2, TREAT3, and TREAT4) exhibited characteristics consistent with normal distribution (Jarque–Bera test, *p* > 0.05), affirming their suitability for parametric analyses.

Following this verification, the selected data analysis methods and their objectives are detailed as follows: Firstly, paired t-tests were executed on CTRL1, TREAT1, CTRL2, TREAT2, CTRL1, CTRL2, TREAT3, and TREAT4 to examine variations in AI scores across different herd sizes and pen numbers. Subsequently, nonlinear regression analysis using the Levenberg–Marquardt algorithm was employed to analyze AI scores across six pig groups at various time points, investigating the relationship between T and AI scores among different groups. Secondly, considering the intra-group factor T, this study utilized sample IDs of four pig groups created in Experiment 1 and four pig groups in Experiment 2, with T as the intra-group variable. Two-way repeated measures ANOVA were conducted with CTRL1, TREAT1, CTRL2, and TREAT2, as well as CTRL1, CTRL2, TREAT3, and TREAT4 as inter-group variables, to explore whether there are differential effects of interaction between T and herd size, as well as the number of pens composing the pig herd. Sphericity tests were performed to assess the assumption of sphericity, with corrections applied using the Huynh–Feldt method when W > 0.75 or the Greenhouse–Geisser correction when W < 0.75. Finally, a conjoint analysis model was utilized to further investigate the primary factors influencing AI scores. Utility values and relative importance of each factor were evaluated to quantify their contributions.

## 3. Results

### 3.1. Differences in Aggression Intensity across Herds

First, we measured the AI of CTRL1, TREAT1, CTRL2, and TREAT2 under different T conditions within their groups. These AIs showed no significant differences between the different pig groups (Appendix A). This indicates that the intensity of aggression towards unfamiliar pigs is not statistically significant between groups of size 9 and size 12 when unfamiliar pigs are integrated into the group.

Subsequently, we measured the AI of CTRL1, TREAT3, CTRL2, and TREAT4 under different T conditions within their groups. These AIs were different (Appendix A). Among them, significant differences were found between CTRL1 and TREAT3 (t = 8.000, *p* < 0.001; Figure 3a), between CTRL1 and TREAT4(t = 7.071, *p* < 0.001; Figure 3b), between CTRL2 and TREAT3 (t = 6.000, *p* = 0.001; Figure 3c), and between CTRL2 and TREAT4, (t = 4.382, *p* = 0.005; Figure 3d), with the mean AI in CTRL1 (3.57) and CTRL2 (3.29) being significantly higher than the mean AI in TREAT3 (2.43) and TREAT4 (2.14). This indicates that there is a statistically significant difference in aggression intensity toward unfamiliar pigs between pig herds composed of two pens and those composed of more than two pens when unfamiliar pigs are integrated into the herd.

### 3.2. Relationship between Herd Establishment Time and Aggression Intensity

To assess the influence of the intra-group factor T on herd aggression intensity toward unfamiliar pigs across various pig herds, and to investigate if this influence is affected by herd size and the number of pens composing the herd, we employed a parabolic model using the Levenberg–Marquardt method for nonlinear regression analysis on T and AI scores across six groups.

The fitting results indicate successful model fitting, with satisfactory goodness of fit metrics (adjusted *R*-square > 0.960; *R* > 0.987; *RMSE* < 0.423; Figure 4; Appendix A). According to the fitting results, the AI showed a significant association with elevated T in six pig herds (Kendall’s *tau-τ* = 0.976, *p* < 0.001). The effect of T on the AI became stronger as T increased, which was consistently validated in six pig herds. These findings demonstrate that as the duration since herd establishment increases, aggression intensity towards unfamiliar pigs significantly escalates within each pig herd in this study.

### 3.3. Interaction Effect between Herd’s Establishment Time and Structure Composition

Considering the impact of the within-group factor T, we conducted repeated measures ANOVA on CTRL1, TREAT1, CTRL2, and TREAT2 to further analyze whether there is a differential effect in the interaction between within-group and between-group factors. The results show that the interaction between CTRL1, TREAT1, CTRL2, TREAT2, and T was not significant (*p* > 0.05; Appendix A). Subsequently, we conducted intra-group effect analysis on CTRL1, CTRL2, TREAT3, and TREAT4. The results show that the interactions of CTRL1, CTRL2, TREAT3, and TREAT4, and T were significant (*p* < 0.05; Appendix A). These findings indicate that there was no significant interaction effect between herd size and T. However, there was an interaction effect between the number of pens used to form herds and T. The interaction effect suggests that there is a significant difference in AI between herds formed with two pens and those with more than two pens when T ≤ 12 (*p* < 0.05; Appendix A). However, as T increased to T > 12, there was no significant effect of the number of pens used to form the herd on the AI.

### 3.4. Contribution Rate and Utility Value of Factors Affecting Aggression Intensity

We performed a conjoint analysis of the effects of T (seven levels), the number of pens (two levels) and size (two levels) on the AI (Appendix A). Their contributions to the AI were 54.660%, 37.783%, and 7.557%, respectively. For T, with T = 12 as the reference, the ranking of utility values for each level was T = 21 (t = 12.211, *p* < 0.001) > T = 18 (t = 8.141, *p* < 0.001) > T = 15 (t = 4.070, *p* < 0.001) > T = 9 (t = −3.256, *p* = 0.003) > T = 6 (t = −8.141, *p* < 0.001) > T = 3 (t = −13.025, *p* < 0.001). For size, the utility of CTRL1 and CTRL 2 was greater than that of TREAT1 and TREAT2 (t = 2.665, *p* = 0.012). For the number of pens, the utility of CTRL1 and CTRL2 was greater than that of TREAT3 (t = −6.092, *p* < 0.001) and TREAT4 (t = −7.833, *p* < 0.001).

## 4. Discussion

Uncovering the factors that influence the intensity of aggression toward unfamiliar pigs is a complex task, but doing so could significantly improve our understanding of the reasons for differences in aggressive behavior across herds. We evaluated whether varying herd establishment times and structural compositions influence the intensity of aggression toward unfamiliar pigs. We showed that differences in the intensity of aggression toward unfamiliar pigs across herds resulted from a complex interaction between herd establishment time and herd structural composition.

### 4.1. Effect of the Structural Composition of Herd on Its Aggression Intensity

This study showed that the AI was not significantly affected by herd size, but it was significantly affected by the number of pens used to form the herd. Although the AI did not significantly differ according to herd size, at 9 ≤ T < 21, the AI differed between herds with sizes of 9 and 12, and the latter was larger. Previous studies showed that group size affects aggression in group-living animals [39], and for pigs, there is less aggression in a group of 12 than in a group of 6 [40]. A possible explanation for why we did not observe similar results for changes in aggressive behavior by herd size as found in previous studies is that there are great differences in the mechanisms of aggressive behavior that occur and that there is variation in different environments [41]. We noted that previous studies demonstrating the relationship between group size and aggressive behavior were performed by simultaneously mixing pigs that were strangers to each other in a new environment. Within the newly formed herd, the pigs were unfamiliar with each other and were uniformly unfamiliar with their environment. In such an environment, since there is no dominant subgroup within the herd at the beginning, aggression manifests itself in one-on-one fights, the purpose of which is not to bite each other to death, but rather to establish a dominant hierarchy to gain access to resources in an orderly manner according to the hierarchical sequence, which means that within a certain range, the larger the group size is, the greater the degree of strangeness within the herd is, and the lower the average competition index among individuals is [42]. However, in this study, the integration of a pig into a new herd environment revealed distinct behavioral patterns. Resident pigs, already familiar with each other and their surroundings, exhibited ‘herd attacks’ towards the integrated pigs initially. Unlike aggression aimed at establishing dominance hierarchies typically seen in unfamiliar pig interactions, these attacks appeared to be driven by curiosity and exploratory behavior among the resident pigs towards the integrated pig [43]. This observation aligns with findings in animal behavior studies highlighting the role of social dynamics, including curiosity, in shaping intergroup interactions [44]. Overall, these differences explain the discrepancy between our findings and those of previous studies on the relationship between herd size and AI, which also reveal that the mechanisms by which aggression occurs are fundamentally different in terms of intrinsic motivation and behavioral consequences between environments, even if they are consistent in terms of behavioral manifestations.

In addition, in actual production, sows undergo herd mixing at least two times throughout their life cycle: at the end of the lactation period into the dry period and at the end of the early stages of gestation into the mid-gestation period [8,9]. For the former case, the sows that end lactation periods at the same time form a new herd by simultaneous mixing, and at this time, the total number of pigs in the herd is equal to the number of pens used to form the herd, and such a herd starts off with no subgroups; thus, the intensity of biting toward sows that integrated late tends to be low. However, for the latter case, sows are typically housed in small groups of four–six during early pregnancy. By mid-pregnancy, the herd size for sows generally increases to 12 individuals or more [45], with even larger sizes if an automated feeding system is used. Sows that move from early gestation into mid-gestation at the same time form a new herd by simultaneous mixing, and at this time, the total number of pigs in the herd is greater than the number of pens used to form the herd; such a herd begins with a dominant subgroup, and thus, the intensity of biting toward any pigs that are integrated late is greater than that of the former case. These phenomena in actual production were explained in this study. This was also the reason for the significant differences in the AI between herds formed with two pens and those with more than two pens.

### 4.2. Effect of the Establishment Time of Herd on Its Aggression Intensity

We found a significant and positive correlation between herd establishment time and aggression intensity, which was consistently observed across different pig herds. At T < 9, the AIs of all six groups were minor and lower, which reveals that the relationships within a group are unstable when a herd is only established for less than nine days, during which time the AI is at its lowest. At T = 18, the AIs of four out of the six groups were severe and above, and at T = 21, the AIs of all six groups were severe and above, and those of four of them were serious, which implies that the time from the establishment to the stabilization of the relationship within a herd is approximately 18–21 days, and it is not appropriate to integrate unfamiliar pigs thereafter. The consistent outward responses of the stable groups to external threats may parallel our findings and those of Morris-Drake et al. (2019) in wild dwarf mongooses and Schaffner et al. (1997) in marmosets [46,47].

In addition, these findings are consistent with actual production, where most pig herds are established by simultaneous mixing for group housing models, during which, in the initial stages of mixing, the pigs are not closely related to each other [20,48]. Hierarchical relationships within the herd are unstable, and minor aggressive behavior occurs from time to time, and it is obvious that during the integration of unfamiliar pigs in this stage, most are not bitten by the resident pigs (T < 9). As T increases, the hierarchical relationship within the herd gradually stabilizes, and AI increases (9 < T < 18, from mild to severe levels). When the hierarchical relationship stabilizes, AI becomes stronger than before (T = 21, serious levels).

### 4.3. Experimental Design and Its Limitations

The selection of time points for the group establishment design was based on considerations of the research application scenarios (the scenarios mentioned in Section 1). The design with a three-day interval was chosen to create a gradient of group establishment times, ranging from 3 to 21 days, to comprehensively assess the impact of group establishment time on inter-group aggression. Additionally, the objective of this study was to explore the effects of group establishment time on aggressive behavior, thus requiring sufficient data support. Adding one pig every three days ensures an adequate sample size within the range of group establishment times for statistical analysis and reliable conclusions. Longer intervals may decrease the number of data points, reducing the effectiveness of statistical analysis and hindering an accurate assessment of the effects of group establishment time. While shorter intervals may lead to pigs being added before the group is fully stabilized, this design allows for a more detailed observation of dynamic processes. This enables capturing changes and adjustments that may occur during the group establishment process, facilitating a more comprehensive understanding of group dynamics and the underlying factors influencing aggressive behavior. In addition to this, there are additional factors to consider. This study encountered highly unique circumstances, which resulted in a series of challenges in experimental design. The current absence of effective control measures to manage the difficulties posed by newly integrated individuals significantly hampers the feasibility of conducting this research in commercial pig farms. Furthermore, national legislation on animal welfare further restricts the scope of our study.

The selection of data analysis methods involved careful consideration of the potential benefits of using mixed models for more precise estimation of variance components and improved interpretation of experimental outcomes. However, significant collinearity issues among our variables were identified during preliminary data analysis. These issues primarily affect the independent variables used in our experimental design, where high intercorrelations could introduce confounding effects and complicate interpretation. Collinearity may complicate model interpretation by increasing interdependencies among variables, potentially biasing estimates of fixed effects and inflating standard errors, thereby impacting the significance testing of model parameters and the accuracy of confidence intervals [49]. Moreover, collinearity can introduce confounding effects, further complicating the interpretation of study results [50,51]. Previous studies suggested strategies to address collinearity, such as removing correlated variables or explicitly specifying variance–covariance structures and dependencies within mixed models [52]. However, based on our preliminary data analysis, these challenges critically affect the accurate estimation of fixed effects in mixed models, potentially compromising their reliability and interpretability. Therefore, we chose repeated measures ANOVA as a more appropriate and robust analytical approach. Repeated measures ANOVA effectively handles within-group factors and provides a clear framework for explaining the variance observed in our experimental design. While mixed models offer flexibility in analyzing hierarchical data structures and repeated measures, our specific data challenges led us to prioritize the application of repeated measures ANOVA.

## 5. Conclusions

This study evaluated whether varying herd establishment times and structural compositions would influence the aggression intensity of resident pigs toward unfamiliar pigs. The AI scores were quantified and assessed using a method based on the severity of skin lesions in integrated pigs. The reasons for differences in AI across herds were analyzed using a parabolic model based on the Levenberg–Marquardt method, and factors influencing AI were evaluated through conjoint analysis. Our results support our hypothesis that T and the structural composition of pig herds significantly affect AI. We also show that the behavioral intensity of resident pigs in biting integrated pigs can be changed by adjusting the T and structural composition of pig herds. Additionally, we found that pigs exhibited exploratory behavior long before biting behavior was expressed during the integration. However, the mechanism by which exploratory behavior transforms into biting behavior and whether biting behavior toward unfamiliar pigs can be avoided by giving resident pigs exploratory results remain open questions that, if answered, could imply the emergence of a new approach to the study of aggressive behavior. Therefore, we will investigate this topic immediately.

## Figures and Tables

**Figure 1 animals-14-02229-f001:**
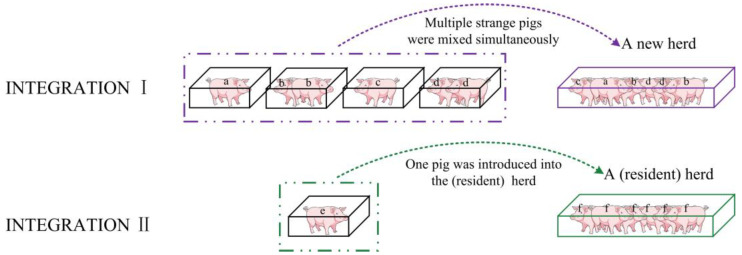
Schematic representation of pig integration. Note that the lowercase letters on the pigs represent pen markings; the same lowercase letter indicates that the pigs originated from the same pen, while different lowercase letters indicate that the pigs originated from different pens.

**Figure 2 animals-14-02229-f002:**
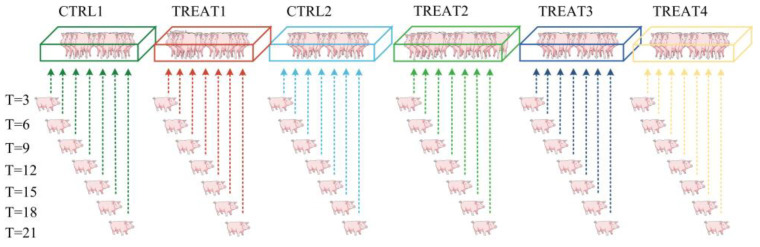
Schematic representation of the integration process in this experiment. Note that CTRL1, TREAT1, CTRL2, and TREAT2 denote the herds of 9 pigs, 12 pigs, 9 pigs, and 12 pigs, respectively. The four groups above originated from two different pens when established. TREAT3 and TREAT4 denote the herds of pigs originally from more than two different pens when established. The herd establishment time was set with 7 gradients, and a pig was integrated into each herd at each time point.

**Figure 3 animals-14-02229-f003:**
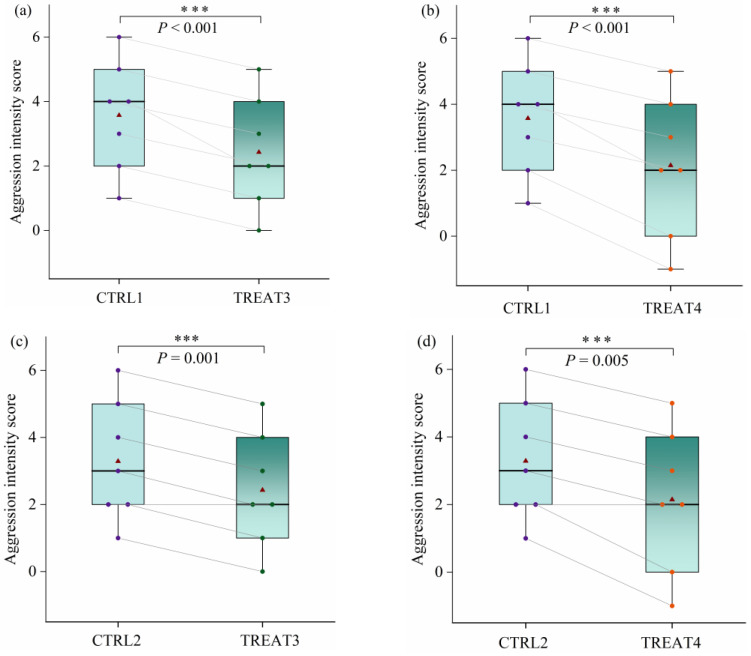
Differences in aggression intensity among the number of pens used to form herds. (**a**) Differences in aggression intensity between CTRL1 and TREAT3; (**b**) differences in aggression intensity between CTRL1 and TREAT4; (**c**) differences in aggression intensity between CTRL2 and TREAT3; and (**d**) differences in aggression intensity between CTRL2 and TREAT4. Note that the thick horizontal black lines in the boxplots represent the median, and the red triangles represent the mean of each group. Asterisks indicate significant differences between the control and treatment groups (*** *p* < 0.01).

**Figure 4 animals-14-02229-f004:**
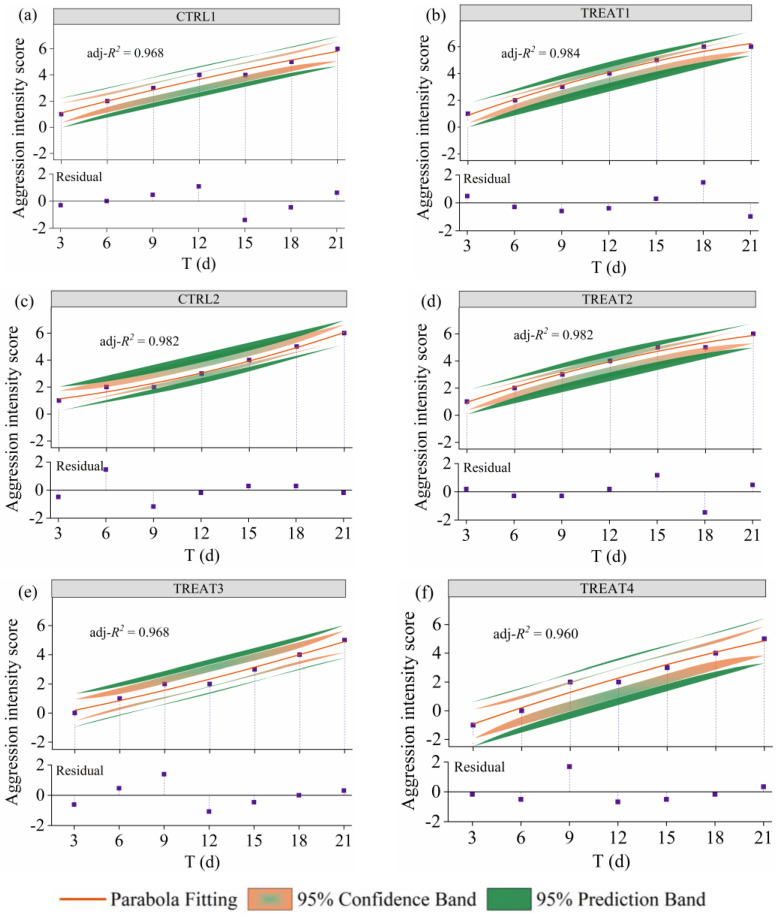
Fitted curve of aggression intensity versus establishment time in the six groups. (**a**–**d**) Fitted curves of aggression intensity vs. establishment time for CTRL1, TREAT1 CTRL2, and TREAT2, respectively; and (**e**,**f**) fitted curves of aggression intensity vs. establishment time for TREAT3 and TREAT4, respectively. Note that the fitted curves, shaded 95% confidence intervals, and prediction bands were calculated from a nonlinear parabola model based on the Levenberg–Marquardt optimization algorithm.

**Table 1 animals-14-02229-t001:** An assignment method for estimating the intensity of aggression toward integrated pigs.

Description for Skin Lesions		Resident Herd	
		Aggression Intensity	Scoring
Integrated pigs	Without skin lesions	/	0
	Slight skin lesions on individual parts	Slight	1
	Minor skin lesions on individual parts	Minor	2
	Multiple minor skin lesions on individual parts	Mild	3
	Moderate skin lesions on individual parts	Moderate	4
	Severe skin lesions across the whole body	Severe	5
	Serious skin lesions across the whole body	Serious	6
Resident pigs	Slight skin lesions on individual parts	/	−1

## Data Availability

Data is contained within the article or Appendix A: The original contributions presented in the study are included in the article/Appendix A, further inquiries can be directed to the corresponding author/s.

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
