# Peer review of "Effects of Herd Establishment Time and Structure on Group-on-Individual Aggression Intensity in Farm Pigs"

_animals, 2024, doi:10.3390/ani14152229_

Round 1

Reviewer 1 Report

Comments and Suggestions for Authors

This study addresses a significant issue in animal husbandry: the aggression in pigs during integration into new groups. This is a critical area of research due to its implications for animal welfare and farm operations. The study is well-designed, the data analysis is thorough, and the findings have notable implications for animal welfare in pig farming. Additionally, this study identifies specific factors and methods that should be considered when mixing pigs in production practice. Overall, the manuscript provides valuable insights into managing aggression in pig herds and enhancing animal welfare. However, there are a few areas where minor revisions or clarifications could further enhance the clarity and effectiveness of the report.

Comments on the Quality of English Language

minor editing 

Reviewer 2 Report

Comments and Suggestions for Authors

This work aimed to investigate the effects of the herd’s composition and the time since the herd was established on aggressions received by a newly integrated individual pig. The research topic is pertinent to the field and potentially has an impact on pig’s behavioral research, especially on finishing pigs and/or sows. However, there are several major concerns that have to be addressed, which are centered around fundamental assumptions and experimental design. See comments below.

Line 14: it should be explicitly elaborated in what context the conclusion was made. In the general pig aggression research area, it has been well established that aggressive behaviors are heritable i.e., genetics contribute to aggressive behaviors in pigs.

Lines 56-59: citations needed that the mentioned scenarios make up a significant portion in practical swine farming so that this study would present value to animal science. In addition, the authors are recommended to justify why the individual pig is integrated into a new social group rather than sent back to the original social group from which it was separated (reference is also needed for this point).

Lines 59-61: citation needed to back up the described behavioral responses.

Lines 62-64: why did the authors assume that the integrated pig would be considered not to be the own kind of resident pigs? If the genetics, litter, and growing environment are the same yet the only difference is social group, would the assumption still hold? In addition, it is described in literature that when pigs are mixed into new social groups, they may re-establish the social hierarchy in the pen. The authors need to justify how the new social group (resident + integrated pigs) is not forming new social compositions. Lastly, the behavioral patterns of pigs may alter significantly throughout the production stage of pigs (e.g., nursery vs finishing/sows). The authors should reflect this point in the context as well.

Materials and Methods: I am not sure why only integrated pigs were recorded instead of the entire social group. Observations from a pilot study wasn’t enough to justify that resident pigs had not been attacked. Reciprocal fights are common aggressive behaviors in pigs. If an integrated pig is attacked frequently, it is possible to fight back. In addition, the authors need to justify why lesion intensity was used in this study, while other traits like aggression duration and lesion counts were not of choices.

The major concern of the experimental design is the setup of Ts. Introducing pigs at earlier times already altered the social group composition, which was different from the original resident pigs. By progressively adding new pigs would also alter the group composition multiple times. Therefore, the marginal effects of time and group composition respectively are not separated in this work but confounded. It would make more sense to introduce integrated pigs only once at each of the six experiment groups at different time points (T=3,6,9,12,15, and 18 for instance), if time is the main effect to be investigated in this study.

Section 2.1: The production stage of pigs needs to be specified and justified.

Lines 192-194: introducing new pigs constantly and frequently into existing groups is already a welfare concern.

Section 2.7.1: I do not understand why two of the dominant analytical methods in the field, social network analysis and linear mixed model approaches, are not adapted in this study. Would an animal scientist prefer algorithms borrowed from engineering/computational methods or established methods in pig aggression studies? For mixed models, the convergence issue can be overcome by using advanced statistical methods such as Bayesian approaches.

Section 2.8: If ANOVA was performed, mixed models should be fitted for more precise estimation of variance components and for better interpretation of the experiment.

Reviewer 3 Report

Comments and Suggestions for Authors

Aggression in farm animals affects welfare. Although one-on-one aggression has been studied, group-on-individual aggression remains unresolved. This study aimed to examine how herd establishment times and structures influence aggression intensity (AI) in pigs.

However, the experimental design of this study: There were so many factors in the experimental design: breed, number of pigs, the number of pens to be mixed, and which day after mixing of a pig to be introduced. But there were only 6 groups in the experiment, and each treatment had only one repetition (pen). It is suggested that the authors reduce the factors analyzed in the experimental design.

The design of mixing a pig into the population every three days is also problematic, because the result of the later time point is influenced by the previous time point. Then the results of each time point cannot be simply compared. The authors should only compare the differences between groups.

The shortcomings of these experimental designs should be discussed in the discussion.

Round 2

Reviewer 2 Report

Comments and Suggestions for Authors

In the revised version, the authors improved the manuscript by adequately addressing my comments. the manuscript was improved significantly and almost ready for publication. However, a few minor comments need to be addressed additionally. See comments below.

1. For Section 2.7.1-2.7.2, the authors should define Jk, muk, rk, d, reduced chi-square, RSS, TSS etc. in mathematical languages and relate the terms to the variables described in Section 2.8. 

In addition, I observed two points that lacked scientific soundness. 

2. the authors mentioned an existing group of familiar pigs would present more aggression to an integrated pig if the latter fights back. This is only observed in the pilot study while the argument has not been scientifically validated or referenced.

3. Which variables had collinearity issues? For statistical analysis, it would be more important to understand what's behind the collinearity instead of choosing an algorithm that may encompass all variables. The challenge mentioned in the manuscript can be dealt with by either removing correlated variables or specifying variance-covariance structure and/or dependent structures of the correlated variables in mixed models. The collinearity issue may be confounding for any algorithms if not taken care of properly. The authors may at least discuss further on the point.

Reviewer 3 Report

Comments and Suggestions for Authors

The authors addressed my concerns after the revision. 

Author Response

Dear Reviewer,

Thank you for your feedback and for reviewing the revised version of our manuscript. We are pleased to hear that the concerns raised have been addressed to your satisfaction. Your guidance has been invaluable in improving the quality of our work.

We appreciate your time and effort in evaluating our submission.

Best regards,

Zhengxiang Shi